# Cytotoxic and Genotoxic Evaluation of Biosynthesized Silver Nanoparticles Using *Moringa oleifera* on MCF-7 and HUVEC Cell Lines

**DOI:** 10.3390/plants11101293

**Published:** 2022-05-12

**Authors:** Hatice Alkan, İbrahim Hakkı Ciğerci, Muhammad Muddassir Ali, Omer Hazman, Recep Liman, Florica Colă, Elena Bonciu

**Affiliations:** 1Faculty of Science and Literature, Molecular Biology and Genetics Department, Afyon Kocatepe University, Afyonkarahisar 03200, Turkey; alkan_hatice_1977@hotmail.com; 2Institute of Biochemistry and Biotechnology, University of Veterinary and Animal Sciences Lahore, Punjab 54000, Pakistan; muddassir.ali@uvas.edu.pk; 3Faculty of Science and Literature, Department of Chemistry, Afyon Kocatepe University, Afyonkarahisar 03200, Turkey; ohazman@aku.edu.tr; 4Faculty of Arts and Sciences, Molecular Biology and Genetics Department, Uşak University, 1 Eylül Campus, Uşak 64300, Turkey; recep.liman@usak.edu.tr; 5Faculty of Agronomy, Department of Agricultural and Forestry Technology, University of Craiova, 13 A.I. Cuza Street, 200585 Craiova, Romania; elena.agro@gmail.com

**Keywords:** DNA damage, nanotechnology, green synthesis, MTT, comet assay

## Abstract

Nowadays, green synthesized nanoparticles (NPs) are extensively investigated to explore their biological potential. They are being explored to treat different infectious and cancerous diseases. Therefore, the current study was designed to evaluate the cytotoxic and genotoxic effects of biosynthesized silver nanoparticles (AgNPs) from the medicinal plant *Moringa oleifera* on breast cancer (MCF-7) and HUVEC (human umbilical vein endothelial cells) cell lines. *M. oleifera*-mediated AgNPs were synthesized from the *M. oleifera* extract (MOE) and then characterized through the use of a scanning electron microscope (SEM), X-ray diffraction (XRD) and UV–vis spectrophotometer. Biosynthesized AgNPs and MOE were employed on MCF-7 and HUVEC cell lines to evaluate their cytotoxic and genotoxic effects. More cytotoxic effects were observed by AgNPs and MOE on MCF-7 cell lines. The IC_50_ for biosynthesized AgNPs was found to be 5 μg/mL. DNA damage was also observed by the MOE and AgNPs on MCF-7 cell lines. However, non-significant DNA damage was observed by MOE and AgNPs on HUVEC cell lines. The findings of the current study revealed the cytotoxic and genotoxic effects of biosynthesized AgNPs on MCF-7 cell lines. However, these AgNPs were considered safe for normal HUVEC cell lines.

## 1. Introduction

Green synthesis is the production of nanoparticles using environmentally suitable materials, such as bacteria, fungi and plants [1]. These green strategies are not just attractive, but also harmless, in comparison to the usual strategies used for synthesis, because they are very friendly to the ecosystem [2]. Several advantages can be obtained by the synthesis of biologically derived extracts, such as a higher production rate and, more importantly, the cheap downstream processing required for the production of the particles [3,4,5]. The synthesis of nanoparticles from plant extracts provides an excellent pathway for the production of nanoparticles that are not only attractive to the industries, but can also be produced on a massive scale. Different studies show that various plant extracts can be used for the formation of silver nanoparticles (AgNPs), having beneficial antimicrobial activities, including extracts of *Acalypha indica* leaves [3], *Solidago altissima* [6], *Xanthium strumerium* L. [3], *Murraya koenigii* (curry leaf) [7], *Ocimum sanctum* (Tulsi leaf) [8,9], extracts from seeds of *Acacia farnesiana* [10], *Macrotyloma uniflorum* [11], extracts obtained from the roots of *Trianthema decandra* [12], stem extracts of *Ocimum sanctum* [8], and even fruit extracts of *Musa paradisiaca* (banana) peels [13] and *Carica papaya* [14]. All these types of studies show that plant extracts possessing phytocompounds can serve as capping or reducing agents in the reaction of silver nitrate (AgNO_3_), which is commonly used to initiate the process of silver nanoparticle synthesis.

*M. oleifera* is an important multitasking crop [15]. The plant is considered to be a rare species because its leaves, seeds, pods and flowers are edible and contain many macro/micronutrients and bioactive substances [16]. Furthermore, the different uses of this plant for livestock (as an important element of feed, fodder and antibiotics), humans (as a food fortificant, cosmetic and cooking oil), and agricultural purposes (purification of water, biofuel and fertilizer) have been published [15,17]. Various parts of *M. oleifera* are commonly used as the major beneficial ingredients in dietary supplement production, because of their broad health-promoting properties [18,19]. The leaves, seeds, flowers and fruits of *M. oleifera* have great nutraceutical and antioxidant potential [15]. The leaves of *Moringa* contain calcium, potassium, iron, protein, vitamins, ascorbic acid, polyphenols, rutin, flavonoids, glycosides and carotenoids, which makes it a superfood for nourishment, and it also has ameliorative effects that combat different carcinogens and mutagens. *M. oleifera* contains biomolecules (such as carbohydrates, proteins and coenzymes) with exemplary potential to reduce metal salt in nanoparticles [20,21]. *M. oleifera* is also used to treat various kinds of illnesses [15], such as for the treatment of menstrual disorders, as a fertility enhancer and cardioprotective, and to reduce abortion [22].

Silver, however, serves some necessary functions as an antiseptic, because of its significant efficiency against about 650 types of disease-causing microorganisms, and it is actually an inorganic and nontoxic agent in its nature [23]. The usage of plant extracts for AgNPs production is not only economic, but also known for its cost effectiveness at the same time [24]. Recent documentations regarding silver plant nano-extracts have shown that it has a vast range of possible applications. Moreover, it has been stated and observed that the leaves of *M. oleifera* are major and primary sources of vitamin C, having a higher quantity of it than that of oranges and lemons, which are considered the main source of this compound [25].

By keeping in mind the great biological potential of *M. oleifera* and vast range of biological activities of silver, the purpose of this study was to control the biosynthesis of *M. oleifera*-mediated AgNPs to synergize their efficacy. Moreover, the chemopreventive properties of *M. oleifera*, through its biosynthesized AgNPs, were also investigated on breast cancer cell lines (Michigan Cancer Foundation-7; MCF-7) and HUVEC (human umbilical vein endothelial cells) cell lines. Both cell lines are important as laboratory model systems. Breast cancer is widespread throughout the world, so it is important to study which agents could be safely utilized for the treatment of breast cancer [25]. Moreover, as HUVECs are cells obtained from the endothelium of veins from the umbilical cord, these cells are normally considered to be excellent cell culture models to study the angiogenesis of endothelial cells after exposure to different mutagens or chemicals [26].

## 2. Results

### 2.1. Biosynthesis and Characterization of AgNPs

The color change from green to brown following the reaction indicated the formation of AgNPs (Figure 1A). The absorption of AgNPs in the range of 350–590 nm, using a UV–visible spectrophotometer, showed a strong and broad surface plasmon peak at 420 nm for biosynthesized AgNPs (Figure 1B). During the formation of AgNPs, changes in the pH value of the mixture were observed. After mixing the two solutions, the pH value measured at the first moment was 6.48, while the pH value at the end of 24 h decreased to 6.09. The surface structure of AgNPs was revealed to have a spherical shape by SEM (Figure 2). Properties such as the crystal structure and grain size of AgNPs were determined by the XRD method (Figure 3).

The obtained Bragg peaks were found to be consistent with crystallographic reflections from 111 (35.68°), 200 (51.62°), 220 (65.86°), and 311 (77.95°), which corresponds to the JCPDS pattern 04-0783. The average size was calculated as 12.09 nm.

### 2.2. Cytotoxic Effects of Biosynthesized AgNPs and Plant Extract on MCF-7 and HUVEC Cells

More cytotoxic effects were observed by both MOE and AgNPs on MCF-7 (Figure 4). The IC50 value was found to be 5 μg/mL for AgNPs. It was also determined that the AgNPs concentrations leading to cell death were cytotoxic at much lower concentrations compared to the plant extract (Figure 4).

Moreover, less cell viability % was observed after 72 h in MCF-7 cells, whereas the cell viability % was not significantly reduced in the HUVEC cell line (Figure 5) by both AgNPs and MOE. Microscopic images of reduced cell viability for both cell lines are shown in Figure 6 and Figure 7. 

### 2.3. Genotoxic Effect of Biosynthesized AgNPs and Plant Extract on MCF-7 and HUVEC Cells

DNA damage was observed by both AgNPs and MOE in the MCF-7 cell line (Table 1) compared to the negative control groups. 

The highest DNA damage (103.67 ± 6.03) was observed at 10 μg/mL in 72 h by the biosynthesized AgNPs. The least DNA damage (48.33 ± 2.52 a) was observed at 1.25 μg/mL in 24 h by the AgNPs in the MCF-7 cell line. This increase in DNA damage was found to be statistically significant. Moreover, similarly, more genotoxic effects were observed by the AgNPs and MOE compared with the positive control. However, MOE and AgNPs demonstrated an almost equal level of DNA damage as the positive control in the HUVEC cell line (Table 2). This indicates non-significant DNA damage in this cell line.

## 3. Discussion

In the current study, biosynthesized AgNPs demonstrated a brown color after their formation. AgNPs exhibit a brown color because of the surface plasmon phenomenon in aqueous solution, which has been previously reported [2,27]. Moreover, the effectiveness of the protocol utilized in the current study is apparent by the fact that the plant secretes carbohydrates, as is observed in *Chorella vulgaris* and in the reaction of AgNO_3_, in which complete reduction takes place at the temperature of 50 °C after 24 h [28]. This, in turn, indicates that there is time variation, which is linked with the type of reducing agent used for the production of nanoparticles. 

The initial results of the microscopic analysis (SEM) showed that AgNPs were spherical. Furthermore, the particles’ appearance was in the form of heavy clusters, which is the result of the physical dehydration that occurred during the procedure of SEM sample preparation [29]. The extraction of *M. oleifera* leaves for AgNPs production in the current study was similar to that of the previous studies [30,31].

Previous studies show the similar consequences of green-synthesized silver nanoparticles in MCF-7 cells [10,32,33]. When there is some alteration in the cells, it has been shown that cell death occurs in the cells, due to the antitumor functions of silver nanoparticles. Similarly, [34] reported the anticancerous assessment of silver nanoparticles against MCF-7 cells. In addition to this, AgNPs can be used in some biochemical functions that enhance the anticancer activities in MCF-7 cells. The authors of [35,36] reported that AgNPs in combination with hyaluronic acid can induce cell death through self-eating, improper functioning of mitochondria, stoppage of cell division, and by inducing peroxidation of lipids.

The comet assay has already been used to assess the DNA damage of various nanoparticles, mutagens, and carcinogens via different model systems [37,38,39,40]. In the current study, genotoxic effects were observed by the biosynthesized AgNPs and MOE in the MCF-7 cell line. Silver nanoparticles have the ability to activate caspase enzymes, along with reactive oxygen species (ROS) that can cause damage to DNA, stress, improper protein folding, and cell death. It has been observed that when activation starts, the cleavage of caspase-3 occurs, and it is translocated through caspase-activated DNA (CAD), due to which DNA fragments are formed. This fragmentation of the nucleic acid by the enzymes is regarded as the main point of natural cell death that might happen at initial steps [41]. A similar observation was reported by [42], who reported the consequences of silver nanoparticles on cellular functions. On the other hand, a predicted mechanism showed that damage to DNA can be due to ROS. ROS are free radicals that are produced in living organisms to perform the normal cellular functions. When there is no appropriate level or amount of ROS present, this decreased level of reactive oxygen species leads to improper functioning of cellular mechanisms, which will induce damage to DNA, peroxidation of fatty acids and cell death [43]. 

However, *M. oleifera* has different biological functions. A variety of these anti-oxidant functions are dependent on the pods, flowers and leaves of seedlings [44]. The initial extracts consist of a variety of mixtures, which have biologically active ingredients. These mixtures can induce cytotoxicity and genotoxicity [45]. Therefore, this property is utilized in the current study to find the anti-cancerous effects through its cytotoxic and genotoxic evaluation via the biosynthesis of *M. oleifera* AgNPs. This study revealed the cytotoxic effects of AgNPs and MOE on both HUVEC and MCF-7 cells. Previously, the cytotoxic effects of *M. oleifera* extracts on human cancer cells, such as pancreatic cancer cells (Panc-1) [46], colon cancer cells (SW480 and HCT18) [47], and KB tumor cells [48], have been reported. AgNPs synthesis using plant extracts is a good alternative to physical and chemical methods, and can be a potential alternative agent in breast cancer treatment [49].

A rise in the activeness of phytoconstituents (total phenolic compounds and total flavonoids) by the use of AgNPs during the process of plant extraction has already been observed [50]. Moreover, the effectiveness of the total antioxidant is increased by the use of AgNPs, and scavenging activity occurs against the free radicals ABTS and DPPH, which increases cytotoxicity, and this is effective against colon cancer cell growth [50,51].

In recent studies, the ability of silver nanoparticles to reduce cell viability and increase apoptosis in different cancer cells has come to the forefront. For this reason, research has led to the selection of natural compounds with anticancer effects for the synthesis of silver nanoparticles, which have valuable properties against cancer cells, but a low effect on normal cells [34,42].

## 4. Materials and Methods

### 4.1. Preparation of Bio-Synthesized Silver Nanoparticles

*M. oleifera* leaves were pulverized and 0.5 gm powder was weighed and mixed with 50 mL of distilled H_2_O. The mixture was heated in a microwave oven (600 W, 50 Hz) for 2 min and the extract was further filtered through Whatman no.1 filter paper. An aqueous solution of AgNO_3_ (1 mM) was prepared and used for the synthesis of AgNPs at room temperature. For the biosynthesis of AgNPs, 980 mL of 1 mM AgNO_3_ solution and 20 mL *M. oleifera* extract (MOE) were mixed. This mixture was heated in a microwave oven (600 W, 50 Hz) for 2 min. The color changed from green to brown, indicating the formation of AgNPs [50].

### 4.2. Characterization of Bio-Synthesized Nanoparticles

After 24 h, the sample taken from the AgNPs solution was examined using a UV–vis spectrophotometer at wavelengths in the range of 320–500 nm. The pH value of the synthesized AgNPs was also measured using a pH meter at the time of creating the first mixture and at the end of 24 h [52,53]. The formation, particle shape and size of the synthesized AgNPs were examined by SEM analysis and the characteristic diffraction of each crystal was analyzed by XRD analysis [54]. Properties such as crystal structure and grain size of AgNPs were determined by the XRD method [50,55]. Characteristic Ag peaks were obtained by takeover XRD spectra of AgNPs synthesized with MOE and 1 mM AgNO_3_ solution at 300 rpm with a magnetic stirrer at 80 °C for 60 min.

### 4.3. Cell Culture

The human breast cancer cell line MCF-7 and HUVEC cell lines were cultured in RPMI-1640 medium containing 10% fetal bovine serum, 100 μg/mL streptomycin, 100 U/mL penicillin and 2 mM l-glutamic acid. Culture was obtained under sterile conditions at 37 °C in an incubator with 5% CO_2_ [56].

### 4.4. MTT Test

Cell viability was evaluated by the 3-(4,5-dimethylthiazol-2-YL)-2,5-diphenyltetrazolium bromide (MTT) test. Briefly, cells were counted and then seeded into 24-well plates. MCF-7 and HUVEC cells were incubated with biosynthesized AgNPs and MOE at different concentrations (1.25–10 µg/mL; 1–100 µg/mL) for 24, 48 and 72 h, respectively. MTT (5 mg/mL) was added (80 μL) to each well and incubated for 4 h. At the end of 4 h, purple-colored formazone crystals formed and those were further dissolved in 800 μL dimethyl sulfoxide (DMSO). Optical density (OD) was measured using a multi-well spectrophotometer (ELISA) plate reader at a wavelength of 540 nm. Concentration (IC_50_ value) of biosynthesized NPs showing 50% reduction in cell viability was determined. Cells untreated with AgNPs were used as the control.

### 4.5. Comet Test

DNA damage was determined by the comet method, as described by [57,58]. Briefly, cells mixed with low-melting agarose were embedded on pre-coated slides of normal-melting agarose. Then, slides were kept in lysis solution and electrophoresis was carried out. This comet-shaped DNA was then stained using ethidium bromide and made visible through a fluorescent microscope. Cells were counted using a fluorescence microscope, as described by [59,60,61].

### 4.6. Statistical Analysis

Results were expressed as the mean ± standard deviation of three independent experiments. The results of the comet analysis were calculated with the Duncan multi-distribution test (*p* < 0.05).

## 5. Conclusions

Biosynthesized NPs are considered to be safer and to have excellent biological potential against cancerous cell lines. Previously, no documented study has been found on the genotoxic and anticancerous assessment of *M. oleifera*-mediated AgNPs on MCF-7 and HUVEC cells, or the comparison between them. It was concluded that biosynthesized AgNPs from MOE exhibited anti-cancerous effects on MCF-7 cells. The MOE extracts also showed cytotoxic and genotoxic effects on MCF-7 cells. Biosynthesized AgNPs and MOE demonstrated that these are considered safe for normal HUVEC cells, as non-significant cytotoxic and DNA damage was observed in this cell line. However, the molecular anti-cancerous mechanism should be further evaluated to better understand its anti-cancerous effects.

## Figures and Tables

**Figure 1 plants-11-01293-f001:**
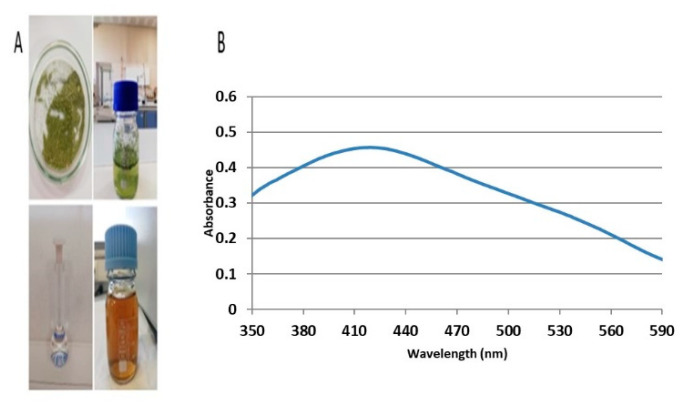
(**A**) Biosynthesis of silver nanoparticles, (**B**) along with UV–visible spectrophotometer image.

**Figure 2 plants-11-01293-f002:**
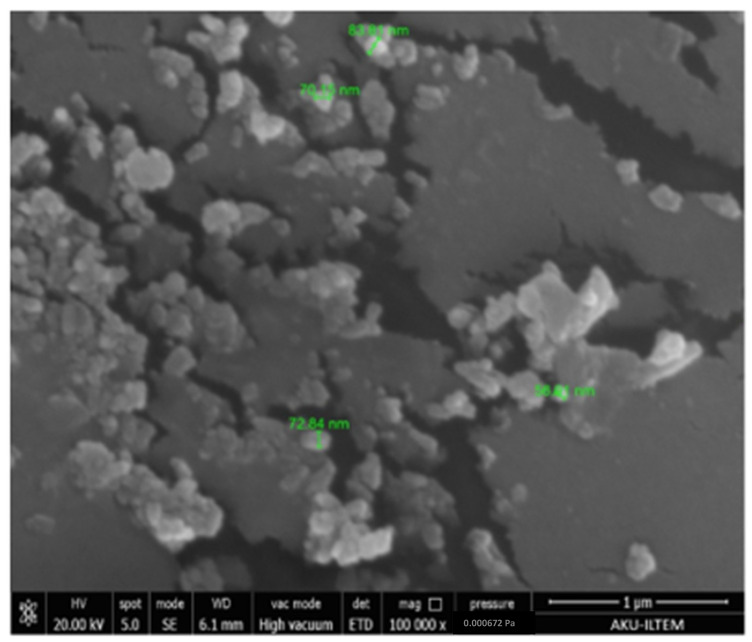
Image of scanning electron microscope (SEM) for the biosynthesized nanoparticles.

**Figure 3 plants-11-01293-f003:**
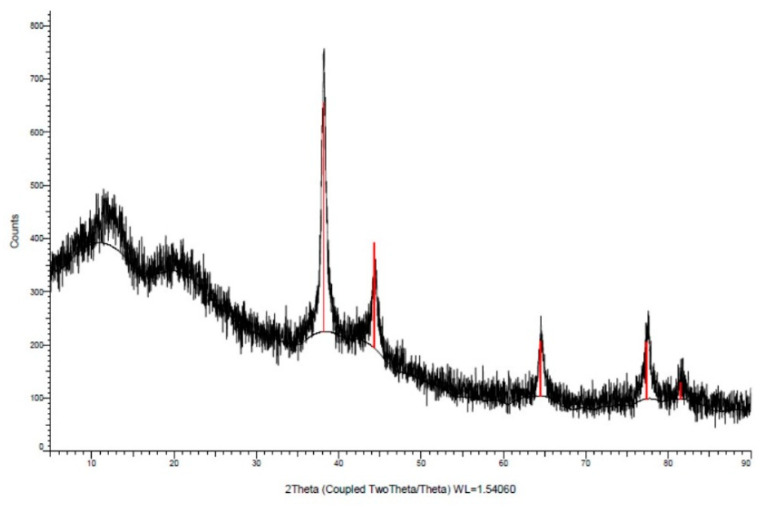
Image of X-ray diffraction for the biosynthesized nanoparticles.

**Figure 4 plants-11-01293-f004:**
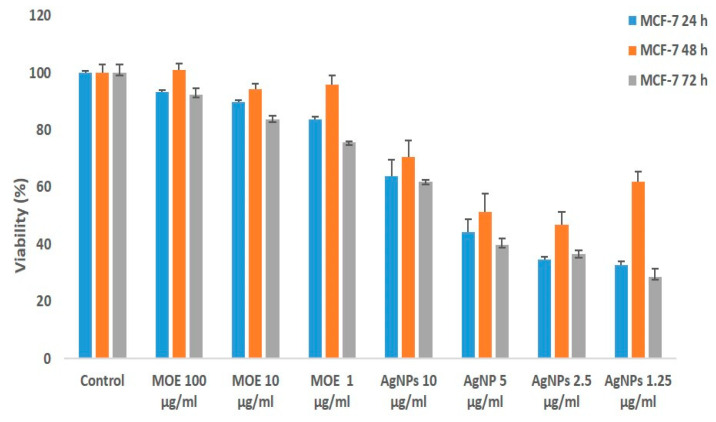
Cytotoxic effect of biosynthesized silver AgNPs and *M. oleifera* extract (MOE) on cell viability of MCF-7. Cells were treated at various concentrations of AgNPs and MOE for 24, 48 and 72 h, and cytotoxicity was determined by MTT. Results are expressed as the mean ± standard deviation of three independent experiments.

**Figure 5 plants-11-01293-f005:**
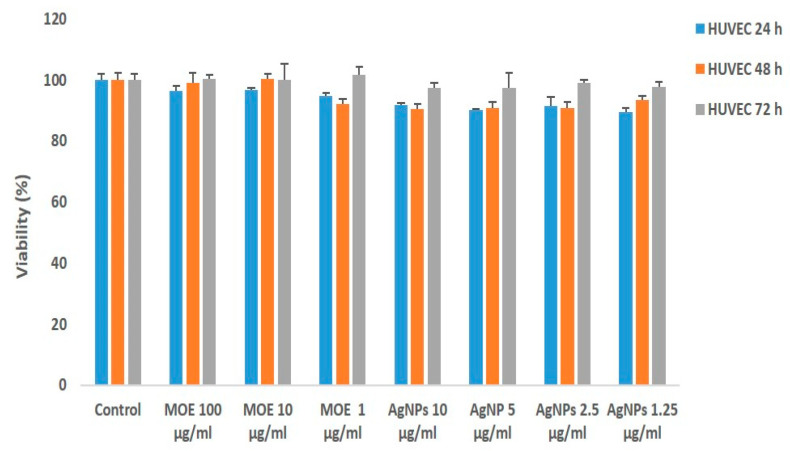
Cytotoxic effect of biosynthesized silver nanoparticles (AgNPs) and *M. oleifera* extract (MOE) on cell viability of HUVEC cell line.

**Figure 6 plants-11-01293-f006:**
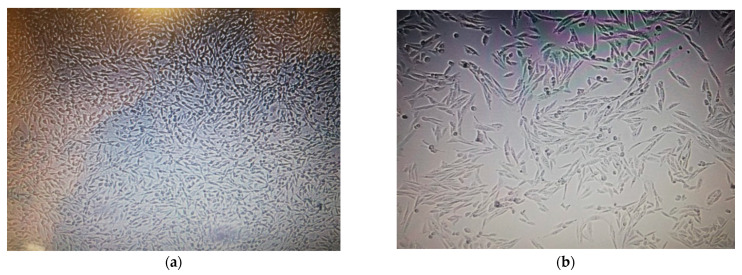
(**a**) Microscopic image of MCF-7 cells after 72 h. MCF-7 cells are well developed and confluent. (**b**) Microscopic image of MCF-7 cells after exposure to NP dosing. Reduced cell viability and less confluence are apparent after dosing.

**Figure 7 plants-11-01293-f007:**
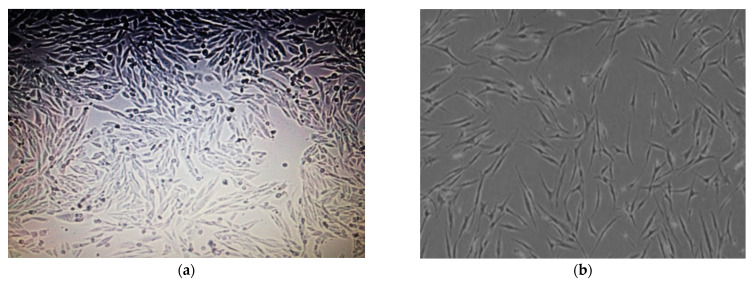
(**a**) Microscopic image of HUVEC cells after 72 h before exposure to NP doses. Cells are well developed and confluent. (**b**) Microscopic image of HUVEC cells after exposure to NP dosing. Reduced cell viability and less confluence are apparent after dosing.

**Table 1 plants-11-01293-t001:** DNA damage at different concentrations of *M. oleifera* extracts (MOE) and biosynthesized AgNPs in MCF-7 cell lines.

Groups	Concentrations (µg/mL)	DNA Damage (Arbitrary Unit) Mean ± Standard Deviation (SD)		
		24 h	48 h	72 h
Control	-	48.67 ± 5.03 a	50.33 ± 5.51 a	53.33 ± 5.86 a
Cisplatin	119	53.67 ± 5.51 ab	63.33 ± 11.55 ab	67.67 ± 12.42 ab
MOE	100	66 ± 4.58 b	107.33 ± 8.08 c	122.67 ± 6.43 f
	10	59.33 ± 8.08 ab	85 ± 11.36 d	99.67 ± 19.4 e
	1	59.67 ± 3.51 ab	85 ± 7.94 d	97 ± 12.12 de
AgNPs	10	61.33 ± 10.07 ab	77.67 ± 4.51 bd	103.67 ± 6.03 ef
	5	52 ± 11.14 a	74.33 ± 7.64 bd	90.33 ± 14.05 cde
	2.5	55.67 ± 9.07 ab	73 ± 5.2 bd	79 ± 9.54 bcd
	1.25	48.33 ± 2.52 a	67.33 ± 2.08 b	71.33 ± 11.02 abc

In each column, means (±SD) with similar letters indicate no significance at *p* ≤ 0.05, depending on the Duncan multi-distribution test.

**Table 2 plants-11-01293-t002:** DNA damage at different concentrations of *M. oleifera* extracts (MOE) and biosynthesized AgNPs in HUVEC cell lines.

Groups	Concentrations (µg/mL)	DNA Damage (Arbitrary Unit) Mean ± Standard Deviation (SD)		
		24 h	48 h	72 h
Control	-	9.67 ± 2.08 a	9 ± 1 a	9 ± 1.73 a
Cisplatin	34	23.17 ± 2.51 ab	33.13 ± 21.12 ab	37.67 ± 11.12 ab
MOE	100	9.33 ± 0.58 a	7 ± 1.73 b	8 ± 1 ab
	10	9.67 ± 1.15 a	7 ± 1 b	7.33 ± 1.53 ab
	1	7.67 ± 2.08 a	6.33 ± 0.58 b	7 ± 1 ab
AgNPs	10	8.67 ± 2.08 a	6.67 ± 0.58 b	7.33 ± 0.58 ab
	5	9.67 ± 2.08 a	7 ± 1 b	6.67 ± 0.58 ab
	2.5	9.33 ± 0.58 a	6.33 ± 1.53 b	6 ± 1 b
	1.25	8.33 ± 0.58 a	6 b	6.33 ± 0.58 b

In each column, means (±SD) with similar letters indicate no significance at *p* ≤ 0.05, depending on the Duncan multi-distribution test.

## Data Availability

Not applicable.

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
