# Peer review of "Cytotoxic and Genotoxic Evaluation of Biosynthesized Silver Nanoparticles Using Moringa oleifera on MCF-7 and HUVEC Cell Lines"

_plants, 2022, doi:10.3390/plants11101293_

Round 1

Reviewer 1 Report

Alkan et alReported Cytotoxic and Genotoxic evaluation of Biosynthesized Silver Nanoparticles using Moringa oleifera on MCF-7 and HUVEC cell lines. Synthesis of nanoparticle via green approach offer advantages. The article is interesting. However, required additional studies for possible publication in Plants journal.

comments:

  1. Scanning electron microscope image requires clarity
  2. Authors need to confirm additional study to determine the size of the particles ..DLS etc.,
  3. The authors studied the cytotoxic effect of synthesised nanoparticles. For better understanding the cytotoxic effect of nanoparticles against cell line images is missing

Author Response

Response to Reviewer 1 Comments

Point 1: Scanning electron microscope image requires clarity.

Response 1: The scanning electron microscope image has been sharpened now.

Point 2: Authors need to confirm additional study to determine the size of the particles .DLS etc.

Response 2: You are right there are a lot of parameters to characterize the Nanoparticles. We performed XRD, SEM and UV Spectrophotometer. Information about particle size is given already in the article and most of the literature states that these parameters are enough to determine the size of the particles.

Point 3: The authors studied the cytotoxic effect of synthesised nanoparticles. For better understanding the cytotoxic effect of nanoparticles against cell line images is missing.

Response 3: Thank you for your valuable suggestion. We have provided now (Figure 6,7 and 8).

Thank you for your time and valuable suggestions.

Yours faithfully,

The Authors

Reviewer 2 Report

Dear authors,

Congratulations for this research, and the interesting results you have obtained. Results are very important, but not the only question to keep in mind. Therefore, I will suggest you some ideas wishing to improve the manuscript.

First, the abstract could be somehow clarified for a better understanding of the reader. There is no reference to breast cancer within the abstract or the intro, and no explanation of MCF-7 and HUVEC meaning. This will of course be obvious for people from the field, but the comparison of the geno and cytotoxic effects between those different cell lines will be easierly understood by other scientists, from the very beginning, and thus the purpose will be clearer. Besides, the word “synthesized” is repeated twice in the first line of the abstract, and one of them might be substituted (produced etc.)

After that, there is a huge mistake when referring to “green synthesis”. It is not the production of nanoparticles using whatever means. In this case, the authors apply a green approach to the production of nanoparticles, but the green chemistry makes reference to a much broader concept, involving the utilization of a set of principles that reduces or eliminates the use or generation of hazardous substances in the design, manufacture and application of chemical products (according to the definition proposed by Anastas and Warner (Green Chemistry: Theory and Practice, P T Anastas and J C Warner, Oxford University Press, Oxford, 1998). Therefore, in this manuscript a green synthesis approach is exploited for both the production of biosynthesized nanoparticles which at the same time might be employed as a green agent for cancer treatment.

Besides, the great biological potential of M. oleifera is cited within the last paragraph of the intro, but it could be described, in order to better understand the choice of this plant, or at least, introduced, leaving some information for the discussion.

Concerning the results, several questions might be improved. First, the bad resolution of the images might be addressed. Besides, the figure 1a could be reshaped into 4 bigger images In line, there is enough room because b is small. Figure 1c displays numbers and words that cannot be read, and the same stands for 1d numbers and axe titles. More uniformity between all the figures would be desirable. Figure 2 huvec 24 h is reflected as “48h”, and mcf-7 and huvec results could be somehow separated within the same image (by color, or zones), to gain more visibility. The text claims for having spherical AgNPs, but it is really difficult to appreciate this within the SEM image. Some lines after that the text says the TEM studies confirm this theory, but there is no TEM image. It would be nice to count on such an image. Anyhow, the shape of these NPs might not be so relevant… Moreover, the authors calculate the average size (I guess by Scherrer equation), and this must be described in more detail. Besides, there is no need to use two decimals, as it is an average size (around 12 nm).

Additionally, it would be nice to count on the explanation of some acronyms (MOE, MTT, the already mentioned MCF-7 and HUVEC, MTT…) to reach a broader audience, and a correlation between the information of light absorbance and the NP size, and that calculated by Scherrer equation might be of interest. Might the broad band in the absorbance spectra be origined by a broad distribution of NP sizes?

Finally, “dose and time dependent an increase…” might be rephrased for the sake of a better understanding. And I might not have understood the results, but in my opinion “dna damage was observed both by AgNPs and MOE IN BOTH CELL LINES” might not be correct: the damage is not so clear for HUVEC cell line, as the effect of the extract and Ag NPs over the dna is under that of control samples, for all the studied concentrations. And that is interesting, because this demonstrates the potential applicability of these extracts as anti-cancer agent, being non-toxic for healthy cell lines, or at least as toxic as the control. However, this is not what the text indicates, and after the explication of results, only dna damage for mcf-7 cell line is explained, an no mention to huvec cell line is made within this part of the manuscript.

Concerning the discussion, TEM image would be nice as already commented. Again, the genotoxic effects for both cell lines are commented, but there are capital differences between them. First introduction of phytoconstituents is made, and I think it should be exploited within the intro. And finally, there is the first reference to properties against cancer cells but low effect on normal cells.

Regarding methods, TEM should be included if the TEM image is also incorporated. And finally, conclusions might be improved, accordingly to the obtained results. Even if the conclusion is very important, the authors might go deep into this question, taking benefit from the discussion ideas.

If these suggestions are duly implemented within the manuscript, I would recommend to accept the article after minor revision.

Author Response

Response to Reviewer 2 Comments

Point 1. First, the abstract could be somehow clarified for a better understanding of the reader. There is no reference to breast cancer within the abstract or the intro, and no explanation of MCF-7 and HUVEC meaning. This will of course be obvious for people from the field, but the comparison of the geno and cytotoxic effects between those different cell lines will be easierly understood by other scientists, from the very beginning, and thus the purpose will be clearer. Besides, the word “synthesized” is repeated twice in the first line of the abstract, and one of them might be substituted (produced etc.) 

Response 1: Thank you for your valuable suggestions. The word ″synthesized″ is replaced with ″produced″. Moreover full form of MCF-7 and HUVEC has been added and in introduction brief literature is also added to highlight the importance of these cell lines in research. Also, ″Nowadays, green synthesized Nanoparticles (NPs) are extensively synthesized to explore their biological potential″ has been changed to ″Nowadays, green synthesized Nanoparticles (NPs) are extensively investigated to explore their biological potential″.

Point 2. After that, there is a huge mistake when referring to “green synthesis”. It is not the production of nanoparticles using whatever means. In this case, the authors apply a green approach to the production of nanoparticles, but the green chemistry makes reference to a much broader concept, involving the utilization of a set of principles that reduces or eliminates the use or generation of hazardous substances in the design, manufacture and application of chemical products (according to the definition proposed by Anastas and Warner (Green Chemistry: Theory and Practice, P T Anastas and J C Warner, Oxford University Press, Oxford, 1998). Therefore, in this manuscript a green synthesis approach is exploited for both the production of biosynthesized nanoparticles which at the same time might be employed as a green agent for cancer treatment.

Response 2: We agree with your comment. Actually, similar kind of synthesis has already been employed to use such kind of green synthesis approach as anti-cancerous agent.  Following are few examples already quoted in the text. But for the safer side we used the term ″Biosynthesized″ instead of green synthesis throughout our article as these nanoparticles were biosynthesized from MOE.

[4] Moodley, J.S.; Krishna, S.B.N.; Pillay, K.; Govender, P. Green synthesis of silver nanoparticles from Moringa oleifera leaf extracts and its antimicrobial potential. Adv. Nat. Sci.: Nanosci. Nanotechnol. 2018, 9, 015011.

[10] Yallappa, S.; Manjanna, J.; Peethambar, S.; Rajeshwara, A.; Satyanarayan, N. Green synthesis of silver nanoparticles using Acacia farnesiana (Sweet Acacia) seed extract under microwave irradiation and their biological assessment. J. Clust. Sci. 2013, 24, 1081–1092.

[11] Vidhu, V.; Aromal, S.A.; Philip, D. Green synthesis of silver nanoparticles using Macrotyloma uniflorum. Spectrochimica Acta Part A: Mol. Biomol. Spectrosc. 2011, 83, 392–397.

Point 3. Besides, the great biological potential of M. oleifera is cited within the last paragraph of the intro, but it could be described, in order to better understand the choice of this plant, or at least, introduced, leaving some information for the discussion.

Response 3: More biological potential of M. oleifera is added briefly in the introduction section now. In Discussion section only relevant literature according to our result has been added.

Point 4. Concerning the results, several questions might be improved. First, the bad resolution of the images might be addressed. Besides, the figure 1a could be reshaped into 4 bigger images In line, there is enough room because b is small. Figure 1c displays numbers and words that cannot be read, and the same stands for 1d numbers and axe titles. More uniformity between all the figures would be desirable. Figure 2 huvec 24 h is reflected as “48h”, and mcf-7 and huvec results could be somehow separated within the same image (by color, or zones), to gain more visibility. The text claims for having spherical AgNPs, but it is really difficult to appreciate this within the SEM image. Some lines after that the text says the TEM studies confirm this theory, but there is no TEM image. It would be nice to count on such an image. Anyhow, the shape of these NPs might not be so relevant… Moreover, the authors calculate the average size (I guess by Scherrer equation), and this must be described in more detail. Besides, there is no need to use two decimals, as it is an average size (around 12 nm). 

Response 4: We have now separated all figures for the better quality of presentation. Now it’s shown as Figure 1, 2 and 3.  And better SEM figure added now to show spherical shape of NPs. Figure is also separated as Figure 4 and Figure 5.

When the crystal sizes are calculated with the strongest peak, which is usually the calculation obtained with this peak is preferred. The average crystal size was found to be between 15 nm and 20 nm. For the calculation, it was calculated by automatically using the Scherrer equation in the program called Diffrac.Eva.V2.1, which is used to determine the mineralogical and crystalline structures and phases of the XRD data.

Point 5. Additionally, it would be nice to count on the explanation of some acronyms (MOE, MTT, the already mentioned MCF-7 and HUVEC, MTT…) to reach a broader audience, and a correlation between the information of light absorbance and the NP size, and that calculated by Scherrer equation might be of interest. Might the broad band in the absorbance spectra be origined by a broad distribution of NP sizes?

Response 5: Those acronyms (MOE, MTT, etc.) are now explained in the article for better understanding. Yes, the broad band in the absorbance spectrum may be due to the wide distribution of NP sizes.

Point 6. Finally, “dose and time dependent an increase…” might be rephrased for the sake of a better understanding. And I might not have understood the results, but in my opinion “dna damage was observed both by AgNPs and MOE IN BOTH CELL LINES” might not be correct: the damage is not so clear for HUVEC cell line, as the effect of the extract and Ag NPs over the dna is under that of control samples, for all the studied concentrations. And that is interesting, because this demonstrates the potential applicability of these extracts as anti-cancer agent, being non-toxic for healthy cell lines, or at least as toxic as the control. However, this is not what the text indicates, and after the explication of results, only dna damage for mcf-7 cell line is explained, an no mention to huvec cell line is made within this part of the manuscript.

Response 6: Thanks for your kind attention on statements. Yes, you are right. We rephrased the sentences accordingly and rewrote this section now. DNA damage was observed by MOE and AgNps in MCF7 cell line whereas the effect of the extract and AgNPs over the DNA is under that of control samples in HUVEC. Detail results in HUVEC are also explained now.

Point 7. Concerning the discussion, TEM image would be nice as already commented. Again, the genotoxic effects for both cell lines are commented, but there are capital differences between them. First introduction of phytoconstituents is made, and I think it should be exploited within the intro. And finally, there is the first reference to properties against cancer cells but low effect on normal cells.

Response 7: Unfortunately, we could not have TEM analysis done in this study because we did not have enough budget and equipment. However, enough characterization has been carried out to see the particle size etc.  Now, we also removed those sentences about TEM from the discussion section. Genotoxic effects for both cell lines are now rephrased in Discussion. As for the details regarding phytoconstituents and their anti-cancerous potential has been discussed, just to relate and strengthen our findings of current study. In introduction we generally gave a lot of details regarding this. But here we specifically discuss with reference to our findings to elaborate the anti-cancerous assessment. Lastly, some relevant references have been placed in the last sentences of Discussion.

Point 8. Regarding methods, TEM should be included if the TEM image is also incorporated. And finally, conclusions might be improved, accordingly to the obtained results. Even if the conclusion is very important, the authors might go deep into this question, taking benefit from the discussion ideas.

Response 8: Unfortunately, we could not have TEM analysis done in this study because we did not have enough budget and equipment. Now we have rewritten the Conclusion section. Thanks for your kind suggestion.

Thank you again for your time and valuable suggestions.

Yours faithfully,

The Authors

Reviewer 3 Report

Nanotechnology has its application in various fields including medicine, agriculture, textile industry and etc. This necessitates the research on various aspects of nanotechnology. Nanoparticles can be synthesized by chemical or physical or biological methods. Green synthesis is a low cost method for the production of nanoparticles and lots of research has been recorded on this aspect. However still many researches are going on to understand the mechanism of green synthesis and other aspects. In the current manuscript, the authors synthesized silver nanoparticles using Moringa oleifera  and evaluated its genotoxic and cytotoxic properties. My comments are,

  1. The abstract needs to be revised well.
  2. What the authors meant by “traditionally grown medicinal plant”? is it traditionally grown medicinal plant or traditional medicinal plant?
  3. Why the authors used MCF-7 and HUVEC cell lines for their studies? A online description on the cell lines in introduction will help the readers to understand.
  4. AgNO3 in line 201 need to changes as AgNO3 Check throughout the manuscript for similar errors.
  5. Results section needs strong revision. For example “The main aspects related to biosynthesis and characterizations of AgNPs are shown 74 in Figure 1.” is not needed. Section 2.2 looks more like materials and methods.
  6. Abbreviations needs to expanded at the first appearance (Eg.SEM; Check throughout the manuscript for similar errors).
  7. Discussion needs a thorough revision.
  8. In discussion authors mentioned “On the other hand, unambiguous proof obtained by TEM analysis of aqueous silver na-147 noparticles indicates that the AgNPs produced were of round shape.” However, results and methods doesn’t support or provide any TEM results. If they refer any other article reference is missing. Make it clear.
  9. Conclusion section needs significant improvement.

Author Response

Response to Reviewer 3 Comments

Point 1: The abstract needs to be revised well.

Response 1: Thank you for your suggestion. The abstract has been revised.

Point 2: What the authors meant by “traditionally grown medicinal plant”? is it traditionally grown medicinal plant or traditional medicinal plant?

Response 2: For a better understanding, the sentence ″So, current study was designed to evaluate the cytotoxic and genotoxic effects of biosynthesized silver (Ag)NPs from the traditionally grown medicinal plant on MCF-7 and HUVEC cell lines″ has been changed to

″So, current study was designed to evaluate the cytotoxic and genotoxic effects of biosynthesized silver (Ag) NPs from the medicinal plant on breast cancer (MCF-7) and human endothelial vein cell (HUVEC) cell lines″.

Point 3: Why the authors used MCF-7 and HUVEC cell lines for their studies? A online description on the cell lines in introduction will help the readers to understand.

Response 3: We have added now in the revised manuscript the necessary explanations and the respective references. Both cell lines are important as a laboratory model system. Breast cancer is prevalent throughout the work so it’s inevitable to study such kind of agents which could be safe and be utilized for the treatment of breast cancer [25]. Moreover, HUVECs are cells obtained from the endothelium of veins from the umbilical cord. These cells are normally considered as excellent cell culture model to study the angiogenesis of endothelial cells after exposure of different mutagens or chemicals [26].

Point 4:  AgNO3 in line 201 need to changes as AgNO3. Check throughout the manuscript for similar errors.

Response 4:  Thank you for your relevant comments. We have edited the entire manuscript accordingly.

Point 5: Results section needs strong revision. For example “The main aspects related to biosynthesis and characterizations of AgNPs are shown 74 in Figure 1.” is not needed. Section 2.2 looks more like materials and methods.

Response 5: According to your suggestions, we rewrote the all results including section 2.2. Also, we removed redundant data of methods. Figures have been improved now. When the crystal sizes are calculated with the strongest peak, which is usually the calculation obtained with this peak is preferred. The average crystal size was found to be between 15 nm and 20 nm. For the calculation, it was calculated by automatically using the Scherre equation in the program called Diffrac.Eva.V2.1, which is used to determine the mineralogical and crystalline structures and phases of the XRD data.

Point 6: Abbreviations needs to expanded at the first appearance (Eg.SEM; Check throughout the manuscript for similar errors).

Response 6: Thank you for your relevant comments. Abbreviations have been reviewed and corrected now.

Point 7: Discussion needs a thorough revision.

Response 7: According to your suggestions, a through revision of Discussion has been made.

Point 8: In discussion authors mentioned “On the other hand, unambiguous proof obtained by TEM analysis of aqueous silver na-147 noparticles indicates that the AgNPs produced were of round shape.” However, results and methods doesn’t support or provide any TEM results. If they refer any other article reference is missing. Make it clear.

Response 8: Thank you for your relevant comments. Yes we agree with you. Now, we have removed these sentences from this section.  Moreover, unfortunately, we could not have TEM analysis done in this study because we did not have enough budget and equipment.

Point 9: Conclusion section needs significant improvement.

Response 9: Thanks for your kind suggestion. The Conclusion section is now revised and improved according to our critical findings.

Thank you for your time and valuable suggestions.

Yours faithfully,

The Authors

Round 2

Reviewer 3 Report

The authors significantly improved the manuscript. However, few more corrections are needed.

  1. Error bar is missing in the graphs
  2. Ligand for Figure 6,7 and 8 may be improved. Also, as per my understanding, MCF-7 cells with and without dosing after 72 hrs has been showed (Fig 6 and 7). But for HUVEC without dosing is not provided.
  3. Conclusion can still be improved by giving the novelty of the study.

Author Response

Response to Reviewer 3 Comments

The authors significantly improved the manuscript. However, few more corrections are needed.

Point 1. Error bar is missing in the graphs

Response 1: Thank you for your suggestion. Now, the error bar it’s added in graphs (Figure 4 and Figure 5).

Point 2. Ligand for Figure 6, 7 and 8 may be improved. Also, as per my understanding, MCF-7 cells with and without dosing after 72 hrs has been showed (Fig 6 and 7). But for HUVEC without dosing is not provided.

Response 2: As you suggested, the ligand for figures is now improved. The presentations of these Figures are now improved as Figure 6 and Figure 7.

Point 3. Conclusion can still be improved by giving the novelty of the study.

Response 3: Thanks for your kind suggestion. The Conclusion section is now revised and improved according to novelty of the study.

Thank you for your time and valuable suggestions.

Yours faithfully,

The Authors
